# Longitudinal Analyses of Mutational Subclonal Architecture and Tumor Subtypes in Recurrent Bladder Cancer

**DOI:** 10.3390/ijms24098418

**Published:** 2023-05-08

**Authors:** Daeun Ryu, Tae-Min Kim, Yun-Hee Lee, U-Syn Ha

**Affiliations:** 1Department of Medical Informatics, College of Medicine, The Catholic University of Korea, Seoul 06591, Republic of Korea; dryu@catholic.ac.kr (D.R.); tmkim@catholic.ac.kr (T.-M.K.); 2Cancer Research Institute, College of Medicine, The Catholic University of Korea, Seoul 06591, Republic of Korea; 3Department of Biomedicine & Health Sciences, Graduate School, The Catholic University of Korea, Seoul 06591, Republic of Korea; 4Department of Urology, College of Medicine, The Catholic University of Korea, Seoul 06591, Republic of Korea; eyh900@catholic.ac.kr

**Keywords:** bladder cancers, longitudinal biopsies, tumor evolution, mutations

## Abstract

Longitudinal tumor sequencing of recurrent bladder cancer (BC) can facilitate the investigation of BC progression-associated genomic and transcriptomic alterations. In this study, we analyzed 18 tumor specimens including distant and locoregional metastases obtained during tumor progression for five BC patients using whole-exome and transcriptome sequencing. Along with the substantial level of intratumoral mutational heterogeneity across the cases, we observed that clonal mutations were enriched with known BC driver genes and apolipoprotein B mRNA editing enzyme, catalytic polypeptide (APOBEC)-associated mutation signatures compared with subclonal mutations, suggesting the genetic makeup for BC tumorigenesis associated with APOBEC deaminase activity was accomplished early in the cancer evolution. Mutation-based phylogenetic analyses also revealed temporal dynamics of mutational clonal architectures in which the number of mutational clones varied along the BC progression and notably was often punctuated by clonal sweeps associated with chemotherapy. The bulk-level transcriptome sequencing revealed frequent subtype switching in which transcriptionally defined BC subtypes may vary during tumor progression. Longitudinal whole-exome and transcriptome sequencing of recurrent BC may advance our understanding into the BC heterogeneity in terms of somatic mutations, cell clones and transcriptome-based tumor subtypes during disease progression.

## 1. Introduction

Bladder cancer (BC) are among the most prevalent human cancer [1]. Over the last several decades, the clinical management of BC has mostly involved surgical resection and conventional chemotherapy. However, the recently developed large-scaled sequencing technologies including genomic and transcriptomic data are expected to considerably change how BC patients are managed. Several studies, including The Cancer Genome Atlas (TCGA) consortium, have identified recurrent mutations as potential drivers of BC genomes, and these findings have aided our understanding into the clinically relevant genomic aspects of BC. For instance, recurrent mutations in cancer-relevant molecular categories in BC were identified, including p53-mediated cell cycle (*TP53* and *CDKN2A/2B*), the RTK-RAS-PI3K pathway (*PIK3CA*) and epigenetic regulators (*KDM6A* and *ARID1A*), for targeted treatments [2,3]. Moreover, the high mutation burden in BC highlights the clinical relevance of immunotherapy to treat metastatic BC [4]. Additionally, it has been noted that the BC genome is predominantly characterized by substitutions contributing to the APOBEC (apolipo-protein B mRNA editing enzyme, catalytic polypeptide)-related signature, and that the APOBEC3G gene also has a role in bladder cancer mutagenesis [5,6]. Non-muscle-invasive BC (NMIBC) and muscle-invasive BC (MIBC) are subtypes of BC with distinct clinical and pathological characteristics and require different treatment approaches. NMIBC accounts for approximately 75% of all BC patients and has a better prognosis compared to MIBC. The treatment for NMIBC typically involves receiving single-agent chemotherapy or intravesical Bacillus Calmette-Guerin (BCG) treatment after transurethral resection of the bladder [7,8]. According to the NIH data updated in February 2023, the 5-year survival rate for NMIBC is 96%, whereas it decreases from 70% for MIBC to 7% for metastatic BC(MBC) [9].

Compared with conventional cross-sectional genomic studies using single biopsies of many patients, studies of genomic intratumoral heterogeneity (ITH) require multiple biopsies, either spatially separated or longitudinal ones, from a single patient. Multiple longitudinal biopsies are often available from individual patients with recurrent diseases, facilitating the analysis of tumor evolution. For example, Faitas et al. analyzed BC genomes of tumors before and after chemotherapy to assess the selective pressure of chemotherapy on the mutation architecture and revealed the early branched evolution for metastatic spread [10].

Transcriptome analysis has provided valuable information regarding tumor intrinsic features (e.g., tumor subtypes) and extrinsic features (e.g., the cellular composition in the tumor microenvironment (TME)). Research has proposed that BC subtypes with distinct biological and clinical natures can be transcriptionally defined [11,12,13]. However, the subtypes have been rarely investigated with respect to temporal or spatial tumor heterogeneity, i.e., the transition of tumor subtypes or co-existence of multiple molecular subtypes in a single BC patient. In addition, given the clinical success of immunotherapy, the TME dynamics associated with tumor progression is worthy of investigation.

In this study, we performed whole-exome sequencing (WES) and RNA sequencing (RNA-seq) for longitudinal specimens obtained from five BC patients. The mutation profiles and subclonal mutation architectures of individual patients were investigated. Mutation-based phylogenetic trajectories were also inferred to trace the origins and evolutionary relationship between specimens including locoregional and distant metastases. We show that the tumor subtyping based on known gene signatures may vary during tumor progression and largely depend on the sites of tissue acquisition, but it is also affected by transcriptome-level of intra-/inter-heterogeneity.

## 2. Results

### 2.1. Clinical History of Five Patients with Recurrent BC

A total of 18 longitudinal surgical specimens were obtained from five BC patients (three to five specimens per case) and subjected to WES and RNA-seq. The clinicopathological information of patients and sequencing statistics are summarized in Table 1 and Appendix A. The schematics of longitudinal acquisition of specimens are illustrated in Figure 1, which shows the time points (months from the first surgery), anatomical sites, classes of NMIBC and MIBC, and types of treatment. For instance, five specimens were obtained from patient 1 (P1) with four surgeries as follows: the first (P1_1; Ta low), the second (P1_2; T1 high), the third and fourth (P1_3 and P1_4 from primary and regional lymph node specimens simultaneously) and the fifth specimen (P1_5 from bone metastasis). The patient received intravesicular administration of BCG and adjuvant chemotherapy after the 2nd and 3rd surgery (annotated as ‘B’ and ‘C’, respectively). For P2, three longitudinal specimens from primary lesions were obtained at intervals of 6 and 13 months, respectively: two specimens (P3_1 and P3_2) were obtained at the initial surgery while the third specimen (P3_3) was acquired from a distant metastatic site. For P4, the initial two specimens, P4_1 (T2 high) and P4_2 (T3 high) were taken at 1-month interval and after 19 and 29 months, the third specimen (P4_3; carcinoma in situ or CIS) and the fourth specimen (P4_4; T1 high) were obtained. A long latency period (60 months) was observed between P5_1 (Ta high) and P5_2 (T1 high) while the third specimen (P5_3, T2 high) was obtained at 94 months from the first surgery. For P1, P4 and P5, NMIBC and MIBC tumors co-existed during tumor progression.

### 2.2. Mutation Landscape of BC Genomes

For mutation-based inference of the evolutionary histories of BC progression, we identified somatic mutations (SNVs and indels) from 18 specimens. In total, 102 to 1300 exonic mutations were observed for five patients (Figure 2A and Appendix A). We also observed that genes frequently altered by somatic mutations and/or copy number alterations in BC [2] were commonly shared across the specimens belonging to the same patient (Appendix A). The level of longitudinal mutational heterogeneity was substantial but variable across the cases. For example, only 11.78% of mutations (74 out of 628 total mutations) were commonly observed in P1, whereas over 50% of mutations were common in P2 and P3. We annotated these common mutations class as ‘clonal’ mutations in contrast to ‘subclonal’ mutations. Up to 59.80% of mutations were classified as clonal mutations including known mutated genes in BC. In P1, *FGFR3* (p.S249C and p.L377R) and *KDM6A* (p.S1114L) missense mutations and one truncating mutation of *KMT2D* (p.E4670*) were observed as clonal mutations, indicating that the mutational makeup for BC tumorigenesis was established early in this patient. Two events of *KMT2D* truncating mutations were observed as clonal-nonsense and subclonal-frameshifting mutations suggestive of the second hit leading to the biallelic inactivation of *KMT2D*. For P2, inactivating mutations of *CDKN1A*, *RB1* and *TP53* were observed as clonal mutations, respectively. For P3, frameshifting indels of *RB1* and *KDM6A* were observed as clonal mutations with nonsense *TP53* mutations. Of note, P4_3 shared only the *MUC16* mutation with other specimens and showed distinct mutation profiles. P4_3 harbored hotspot missense mutations of *PIK3CA* (p.H1047R) as well as *KMT2D* and *ERCC2* missense mutations not shared by the other specimens suggestive of a distinct clonal history. Also, P4_3 with a distinct histologic presentation of CIS may present a tumor with a metachronous origin. Therefore, mutations shared by P4_1, P4_2 and P4_4 were considered as clonal mutations, which included a frameshifting indel of *CDKN1A*. For P5, the extent of mutational overlap of the early specimen (P5_1) with the remaining specimens (P5_2 and P5_3) was suggestive of early clonal branching. Of note, we observed that early and later lesions harbor two distinct *ERBB2* missense mutations (p.E332K in P5_1 and p.S310Y in P5_2/P5_3), which is suggestive of functionally converging *ERBB2* mutations.

The clonal and subclonal mutations were investigated for their relative frequencies of six mutation spectra and changes in amino acid residues (Appendix A), but no substantial differences were observed between clonal and subclonal mutations. We further analyzed the mutation signatures of clonal and subclonal mutations (Figure 2B). Mutation signatures of single base substitution 1 (SBS1) representing clock-like mutations were frequent both for clonal and subclonal mutations, suggesting that the mutation abundance of BC genomes can serve as molecular clocks [14]. Of note, mutations related to the activity of spontaneous APOBEC deaminase (SBS2 and SBS13) were more frequent in clonal mutations than sub-clonal mutations, suggesting that the early mutation makeup for BC tumorigenesis may have been largely attributed to APOBEC mutagenic activities (*p*-value = 0.05216, *t*-test). Variant allele frequencies were significantly higher for clonal mutations than those of subclonal mutations (*p*-value = 1.31 × 10^−5^, *t*-test, Figure 2C).

### 2.3. SCNA Profiles of BC Progression

Genome-wide SCNA profiles are shown to illustrate the arm-level and focal/gene-level gains or losses for individual specimens (Figure 3). Despite the overall concordance of SCNAs across specimens within cases, the gene-level SCNA profiles for known BC drivers showed a substantial level of intra-patient heterogeneity. In the case of P1, gains of 1q and 8q were shared by all specimens except for P1_5, and the deletion of *CDKN2A*, which was proposed as a marker related to the progression of MIBC in NMIBC [15], was observed in all specimens except for P1_3. In P2, the gain of *NECTIN4*, *E2F3* and loss of *PTEN* were observed as clonal SCNAs. P3_3 showed distinct gene-level SCNA profiles suggesting that P3_1 and P3_2 may have occurred after the divergence of clones for P3_3. Given the relatively high level of mutational concordance for this case, this is suggestive of the late-occurring nature of SCNA in BC evolution. While the SCNA profiles of two early specimens (P4_1 and P4_2) were largely concordant, P4_3, proposed as a metachronous lesion according to the mutation profile and P4_4, was distinct from P4_1 and P4_2 except for shared deletions such as those including *FGFR3*. For P5, P5_3 showed several chromosomal gains, some of which were consistent with those of P5_1, but a relative lack of SCNA concordant with P5_2. Discrepancies in terms of both arm- and gene-level SCNA profiles, also consistent with the mutation profiles, may reflect the clinical history of the patient.

### 2.4. Cancer Evolution Inferred from the Multiregional Mutation Profiles

We employed CALDER [16] to infer the mutation-based phylogenies using longitudinal sequencing data. The tumor evolution was reconstructed as temporal changes in the abundance of cell clones (annotated as capital letters, in which A represents the ancestral clone) and defined as population of cells harboring a set of distinctive mutations (annotated from M1 in order of acquisition). The prevalence of cell clones (7–11 clones per patient) and their evolutionary relationships are presented as phylogenetic trees across the longitudinal axis in Figure 4. The constructed phylogenies represent the clonal dynamics across tumor progression. Clone A of P1 represents the ancestral clone occurring early in the cancer evolution and persists throughout the evolution. Of note, clone A remained as the sole clone in the last specimen (P1_5) obtained after the chemotherapy, indicative of an evolutionary bottleneck to which the clone D and its derivatives of clone I, J and K were subject. Clone A possessing stem-like properties may have contributed to several progeny clones during tumor progression and survived after the chemotherapy. For P2, the level of heterogeneity declined during the tumor progression, i.e., five clones are present in P2_1 while three and two clones survived in P2_2 and P2_3, respectively. P3 showed clonal transition, in which a dominant clone (clone D) of P3_1 is replaced by pre-existing clones of B and E. The evolutionary bottleneck associated with the chemotherapy leaves clone G as the sole cell population remaining at P3_3. Mutational clonal architectures were similar for specimens acquired during the same surgery, such as P4_1 and P4_2. For P4, persisting clones (clone A and C) served as source for newly arising clones with the acquisition of mutations (e.g., clone F with *ERCC2* mutations) during tumor progression. The clonal sweep was observed for P5_2 where the diverse clonal presentation of P5_1 was replaced by newly arising clone F.

### 2.5. BC Subtypes along Tumor Progression

Gene expression may stratify BC patients with distinct biological features and clinical outcomes. Several classification schemes have emerged [11,12,13] and consensus BC subtypes (LumP, LumNS, LumU, stroma-rich, Ba/Sq, and NE-like) have been recently proposed [17]. Figure 5A shows consensus subtypes across 17 specimens and the expression of genes in seven molecular categories. Appendix A also summarizes molecular subtypes made by different sub-classification systems in comparison with six consensus subtypes. P2 with three primary specimens was constantly annotated as the luminal unstable (LumU) subtype. For P1, the initial two primary and lymph node specimens (P1_1, P1_2 and P1_4) were annotated as luminal papillary (LumP) but P1_3, was classified neuroendocrine-like (NE-like). Distant metastatic specimens such as P1_5 (bone metastatic lesion) and P3_3 (abdominal wall metastasis) were annotated as stroma-rich, which may be attributed to the high level of infiltrating stromal cells in distant metastatic sites. Tumor subtyping of P4 and P5 were not consistent due to the high level of expression of extracellular matrix (ECM)- or epithelial-to-mesenchymal transformation (EMT)-related genes despite the overall expression of luminal marker genes. We further examined the abundance of tumor infiltrating immune and stromal cells in the TME and the activity of BC-related molecular gene signatures [18,19] (Figure 5B). Luminal subtypes were relatively enriched for neutrophils and endothelial cells, and NE-like and stroma-rich subtypes were enriched with NK cells and cytotoxic lymphocytes. The high level of infiltrating immune cells may be attributed to the tissue characteristics such as local lymph nodes. We also observed that luminal subtypes consistently showed up-regulation of urothelial differentiation. The FGFR co-expressed genes and cell cycle genes were relatively up-regulated in LumP and LumU subtypes, respectively. While the NE-like subtype up-regulated the neuroendocrine differentiation as well as basal differentiation, keratinization signatures were relatively up-regulated in stroma-rich tumors.

## 3. Materials and Methods

### 3.1. Patients

Five patients who experienced recurrence and underwent surgery repeatedly after the first diagnosis for bladder cancer were enrolled for this study. This study protocol was approved by the Institutional Review Board of Seoul St. Mary’s Hospital, College of Medicine, the Catholic University of Korea obtained (approval number: KC19SESI0223). Blood samples were obtained from all patients, and written informed consent was obtained from all participants.

### 3.2. WES and RNA-Seq and Pre-Processing

To capture exonic DNA from formalin-fixed paraffin embedded tumor tissue and matched blood samples, we used SureSelect Human All Exon V6+UTR (Agilent, Santa Clara, CA, USA) according to the manufacturer’s protocol. The 101 bp–sequencing reads were aligned to the hg19 reference and further processed using the Genome Analysis ToolKit (GATK, v4.1.7.0) [20] in accordance with the best practices pipeline [21]. For RNA-seq, we used the SureSelectXT RNA Direct Reagent Kit (Agilent, USA) to construct sequencing libraries. The pre-processing of RNA-sequencing data were mapped to the GRCh38 human reference and conducted according to the National Cancer Institute (NCI) Genomic Data Commons workflow [22]. For quality control of sequencing, we performed qualimap (v2.2.1) with the aligned bam files [23].

### 3.3. Genomic Analyses

According to the GATK pipeline, we used Mutect2 to identify somatic mutations by comparing the tumor and matched normal sequencing data and annotated variants with Funcotator. To ensure the sensitive mutation calling, we leveraged the presence of multiregion samples as previously described [24,25]. We used deconstructSigs (v1.8.0) [26] and COSMIC mutation signature ver. 2 (https://cancer.sanger.ac.uk/signatures/signatures_v2/, accessed on 20 November 2020) to estimate the abundance of mutation signatures. For joint calling of somatic copy number alterations (SCNA) across specimens in each case, all sample-level breakpoints were obtained from the CNVkit (v.0.9.8) [27]. VarScan2 (v2.4.3) [28] was used to calculate GC-corrected tumor-vs.-normal copy number ratios in 1 kb bins, which were summarized into the sample-level segmentation profiles using the patient-level breakpoints integrated per patient. To identify the genetically distinct cell populations and to estimate the temporal changes in clonal abundances along the tumor progression, we used CALDER (v0.11) [29]. To estimate the mutation frequencies, we used the cancer cell fraction as estimated by ABSOLUTE (v.1.0.6) [29] to exploit the mutations in copy number non-neutral regions.

### 3.4. Transcriptomic Analyses

The tumor subtyping based on gene expression was done using the R package (https://github.com/cit-bioinfo/BLCAsubtyping, accessed on 4 March 2021) that annotates the six consensus subtypes of individual cases [17]. MCPcounter (https://github.com/ebecht/MCPcounter, accessed on 4 March 2021) [30] was used to estimate the abundance of immune and stromal cells in the TME. Gene set variation analysis [31] was performed to estimate the level of enrichment for BC-related gene signatures.

## 4. Discussion

Somatic mutations have served as evolutionary markers to delineate the evolutionary dynamics of cancer genomes. While the single-time biopsy-based snapshot of mutation profiles has been a major resource for cancer studies and clinics, the longitudinal genetic studies leveraging the temporally acquired biopsies can provide insights into both the evolutionary dynamics of cancer genomes and potential guide to clinical decisions. The early studies of ITH including metastatic lesions showed the distant metastases may resemble the genetic makeup of their originating primary tumors with a striking concordance of cancer drivers between primary and matched metastatic lesions [32]. However, metastatic lesions can acquire a substantial number of genetic alterations from the divergence [33] and the dynamics of genomic evolutionary may be accelerated with cancer treatments. A recent comprehensive multi-region mutation profiling of lung cancers by TRACERx has further demonstrated the complex origins of metastatic clones with a potential limitation of current radiologic screening [34].

This study aimed to investigate the evolutionary dynamics of BC progression by analyzing longitudinal samples obtained from patients. The research focuses on three aspects of the molecular heterogeneity associated with BC progression. First, specimens obtained at disease recurrence showed a considerable mutational heterogeneity (“inter-tumoral heterogeneity”). The mutational heterogeneity contributes to the drug resistance and disease recurrence affecting the clinical outcomes of BC patients [35,36]. Although the level of heterogeneity was variable across the observed cases (“inter-patient heterogeneity”), clonal mutations consistently showed the enrichment of BC driver genes and APOBEC-related mutation signatures, suggesting that mutation forces operative during the early and later BC evolution are distinguished. In spite of the heterogeneity, the majority of BC driver mutations were observed as clonal mutations. For example, patients with *FGFR3* mutations, mostly observed as clonal mutations, may be eligible for the recently approved erdafitinib [37]. However, the dynamics of mutational architecture as well as a substantial level of mutational heterogeneity should be taken into account. We observed that SCNAs may represent later genomic events occurring after the divergence of branching evolution for some cases [35].

Second, the phylogenetic tree representing the mutational subclonal architectures and their evolutionary relationship supported that the initial tumor specimens already include multiple cell clones whose abundances vary along the tumor progression. The proliferation or depletion of individual clones depends on the evolutionary fitness conferred on the clones. The fitness is also dependent on the cellular context such as during chemotherapy and we observed that chemotherapy reduced the number of cell clones representing the evolutionary bottleneck. In P1, we observed that the most ancestral clone (P1-A) produced its progenies during tumor progression and survived when the final specimen was obtained. For P5, the most ancestral clone (P5-A) produced its progenies clone until tumor relapse, suggesting these clones possess stem cell-like behaviors persisting throughout the tumor life cycle, giving rise to multiple cell populations also resistant to chemotherapy.

Third, the transcriptome-based BC classification was often discordant during the disease progression or according to the tumor sites. The discordance of subtypes in serial samples or ‘subtype switching’ should be taken into account for transcriptome-based molecular diagnostics of patients. Subtype switching has been observed for cases with the progression of NMIBC to MIBC (e.g., P1 and P5). Interestingly, P2 without the progression to MIBC also showed concordance of tumor subtypes. The discordance of subtypes along the tumor progression may indicate the presence of undiscovered subtypes or the co-existence of multiple subtypes [38]. BC subtype switching as well as their transition during disease progression can be further explored using high resolution single cell RNA-seq for individual cell–level subtyping.

## 5. Conclusions

The current BC management is largely based on pathologic features and tumor staging, but the current system is relatively limited to cover the extensive heterogeneity of tumors for optimal disease monitoring and clinical decision. Our findings demonstrate the substantial level of genomic and transcriptomic heterogeneity of BCs during tumor progression, and these features should be taken into account for the clinical management of the disease. Although we obtained serial specimens during the repeated disease recurrences, no common genetic alterations across patients were observed, largely due to the substantial inter-patient heterogeneity and small case numbers. Taken together, our findings highlight the need for longitudinal acquisitions for better BC management, which can be facilitated by technical advances such as evaluation of circulating DNA by liquid biopsies [39].

## Figures and Tables

**Figure 1 ijms-24-08418-f001:**
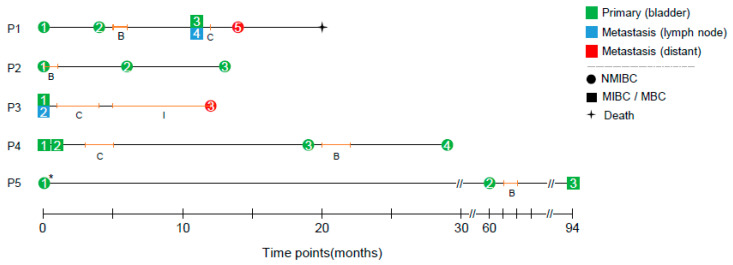
Clinical history of five bladder cancer (BC) patients. The clinical history including the surgical acquisition of tumor specimens and the treatments of individual patients (P1–P5) are depicted on the time scales, representing months passed from the initial surgery. Tumor specimens were distinguished into primary tumors (bladder) and metastases (regional lymph nodes or distant). NMIBC and MIBC represent non-muscle-invasive BC and muscle-invasive BC (presented as circle and square, respectively). The duration of treatment is indicated by an orange line with the type of the treatment (B: BCG, C: chemotherapy, I: immune therapy). The specimens were analyzed by whole-exome and transcriptome sequencing with one exception indicated by an asterisk (only whole-exome sequencing was available for the case).

**Figure 2 ijms-24-08418-f002:**
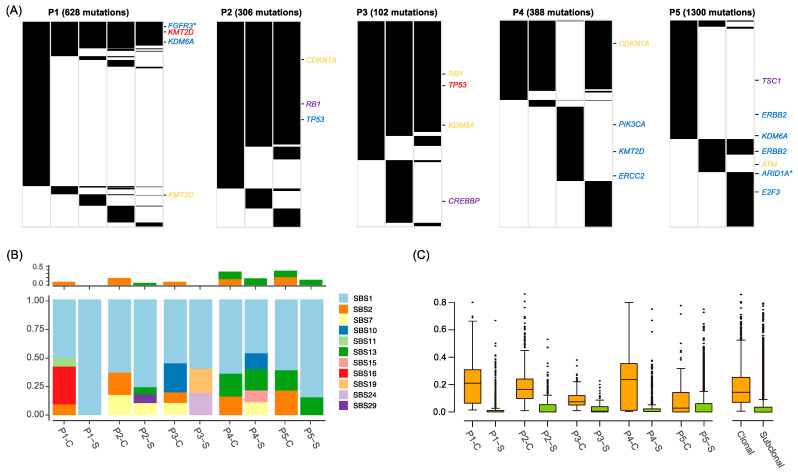
Somatic mutations of BC genomes. (**A**) The landscape of somatic mutations (SNVs and indels) along the disease progression is illustrated by the presence(black) or absence(white) of mutations in each specimen. Mutations and the associated genes previously reported to be recurrent and associated with BC pathogenesis are shown (blue: missense, red: nonsense, yellow: splice-site, purple: frame shift in/del). * indicate two mutation sites (resulting in different protein changes) within the gene. (**B**) The relative fraction of mutation signatures (SBS, single based substitutions) is shown for clonal and subclonal mutations in individual patients (C, clonal; S, subclonal). The APOBEC-related mutation signatures of SBS2 and SBS13 are separately shown for their abundance (above). (**C**) Boxplot represents variant allele frequency for clonal (orange) and subclonal (green) mutations in each patient.

**Figure 3 ijms-24-08418-f003:**
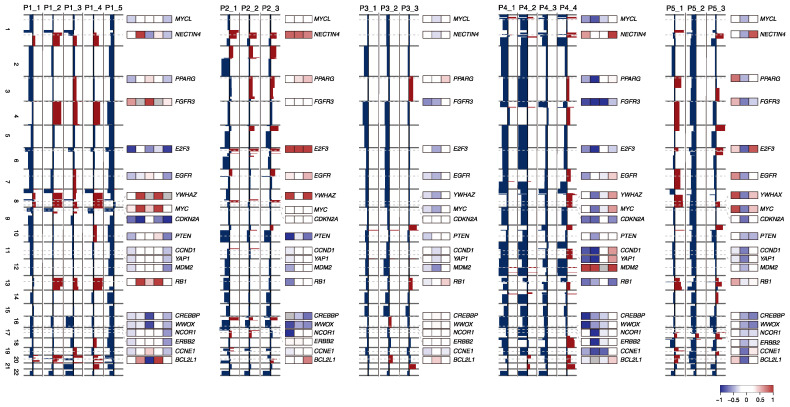
Genome-wide SCNAs profiles of BC genomes. For P1–P5, genome-wide SCNA profiles showing arm-level copy number alterations are illustrated (red and blue as copy number gains and losses, respectively). The gene-level copy number log2ratios are shown as those of their belonging copy number segments at the corresponding loci. The threshold for genome-wide SCNA is +0.25 and −0.25 for gains and losses. Color indicates the level of copy number alterations for gene-level SCNAs.

**Figure 4 ijms-24-08418-f004:**
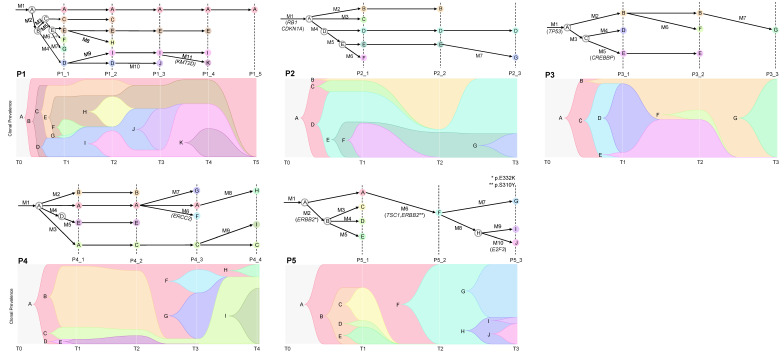
Mutational clone architecture and evolutionary trajectory. For each patient, mutational clones are defined as cell populations with distinct mutational makeup (A to K, in order of appearance in the tumor evolution for P1). The phylogeny represents the evolutionary relationship between clones and sets of mutations acquired during the transition between cell clones are indicated by M (M1 to M11). Mutations of known BC drivers are denoted (top). The clonal prevalence, the relative fraction of individual cell clones in each specimen estimated by matrix factorization, is presented along the tumor progression or the temporal order of individual specimens (bottom). The arrow represents the phylogeny of the mutation clone, and each clone has a different color.

**Figure 5 ijms-24-08418-f005:**
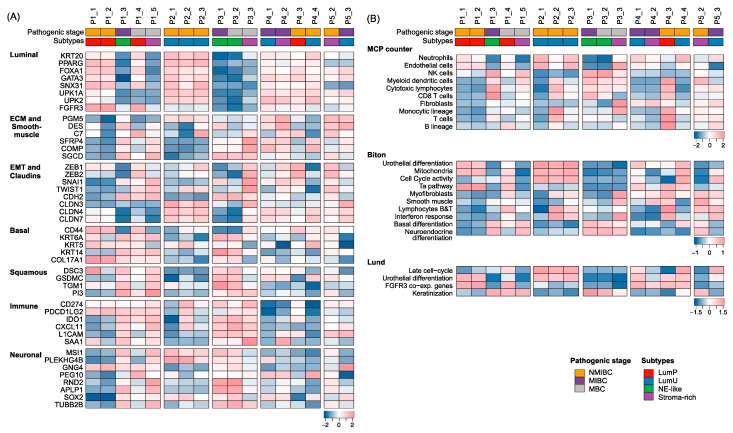
Transcriptome profiling for BC–related marker genes and signatures. Hierarchical clustering of expression heatmaps showing expressed subtype marker genes (**A**), immune profiling and gene signatures (**B**).

**Table 1 ijms-24-08418-t001:** Clinical history of bladder cancer patients.

		P1	P2	P3	P4	P5
Age		67	63	68	63	78
Sex		Male	Male	Male	Male	Male
Number of samples		5	3	3	4	3 (2 *)
1st	Tumor Stage	Ta	T1	T3aN2/MBC (L)	T2	Ta
	Tumor Grade	Low	High	High	High	High
2nd	Tumor Stage	T1	T1	MBC (D)	T3	T1
	Tumor Grade	High	High	High	High	High
3rd	Tumor Stage	T3bN+/MBC (L)	T1		CIS ^+^	T2
	Tumor Grade	High	High		High	High
4th	Tumor Stage	MBC (D)			T1	
	Tumor Grade	High			High	
Treatment						
BCG		1	1		1	1
Chemotherapy		1		1	1	

MBC: metastatic bladder cancer (Lymph—L, Distant—D). * Number of RNA sequencing samples. ^+^ Carcinoma in situ.

## Data Availability

The data presented in this study are available in Appendix A.

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
