# Peer review of "Longitudinal Analyses of Mutational Subclonal Architecture and Tumor Subtypes in Recurrent Bladder Cancer"

_ijms, 2023, doi:10.3390/ijms24098418_

Round 1

Reviewer 1 Report

In this work, the authors performed whole-exome sequencing (WES) and RNA sequencing 62 (RNA-seq) for longitudinal specimens obtained from five BC patients. the authors used the mutation profiles and subclonal mutation architectures of individual patients. the author showed that the tumor subtyping based on known gene signatures may vary during tumor progression and largely depend on the sites of tissue acquisition, but it is also affected by transcriptome-level of intra-/inter-heterogeneity. I have gone through the manuscript, and I found the topic and the work done of great interest, and suitable for publication in “International Journal of Molecular Sciences”. The work presented is diversified and includes many important results. I recommended the manuscript for publication in “International Journal of Molecular Sciences”

Reviewer 2 Report

This is a very well written and impactful manuscript. I have the following concerns and comments:

1.       Abstract, Introduction: Please define the APOBEC acronym, I also suggest the following edit ‘…APOBEC-associated gene mutation signatures…’ – not all readers may be familiar with this gene family. I also suggest including more information about this gene family including some details/citations relating to their role in driving bladder carcinogenesis.

2.       Introduction: The authors should provide some background explaining the difference between non-muscle invasive and muscle invasive bladder cancers as tumor samples from both are collected and analyzed in this study. These have very different genetic profiles and incidence and survival rates, and the treatments are also very different – including this information will help readers better understand the value of the data collected in this study.

3.       Table 1: typo says ‘chemotheapy’ instead of ‘chemotherapy’

4.       Table 1: ‘N’ and ‘E’ (smoking history) as well as ‘M’ and ‘F’ (Sex) should be defined in the footnotes section.

5.       Discussion: I encourage the readers to further expand on how genetic studies, and in particular longitudinal genetic studies, can be used to help guide treatment decisions as well as challenges involved. Please also provide examples of targeted the

6.       Results, Discussion: Please comment on any findings relating to the changes (if any) in the expression levels of targets of immunotherapies which can be used to treat bladder cancers, e.g. PD-1 and PDL-1.

Reviewer 3 Report

The authors presented evidence showing that substantial level of genomic and transcriptional heterogeneity of BCs during tumor progression. It is an interesting study. Overall, I think the writing and findings are very good. I have only a few minor comments.

1, It would be great if more recently published references are used in the article, especially in the introduction section.

2, Indeed, as the authors mentioned, the sample number was too small. However, this is something that you may not be able to control. So if the conclusion or concept in BC by this study can be also supported by other publications using different cancer models, it may be interesting to discuss in the last section.
